# The Role of Exhaled Breath Analyses in Interstitial Lung Disease

**DOI:** 10.3390/diagnostics15222884

**Published:** 2025-11-14

**Authors:** Panaiotis Finamore, Alessio Marinelli, Simone Scarlata, Silvano Dragonieri, Andras Bikov

**Affiliations:** 1Unit of Internal Medicine, Fondazione Policlinico Universitario Campus Bio-Medico, 00128 Roma, Italy; 2Department of Respiratory Diseases, University of Bari, 70121 Bari, Italy; 3Wythenshawe Hospital, Manchester University NHS Foundation Trust, Manchester Academic Health Science Centre, Manchester M23 9LT, UK; 4Division of Infection, Immunity and Respiratory Medicine, Faculty of Biology, Medicine and Health, The University of Manchester, Manchester M13 9PL, UK

**Keywords:** ILD, VOC, EBC, breathprint

## Abstract

Interstitial lung diseases (ILDs) represent a group of lung disorders that primarily affect the lung parenchyma. These disorders are usually progressive, may be debilitating and life threatening, and often pose diagnostic and therapeutic challenges. Exhaled breath analyses offer opportunity for diagnosis, differential diagnosis, and to predict prognosis and treatment outcomes. Numerous studies have been published using various exhaled biomarker analyses, including exhaled nitric oxide, exhaled breath condensate, and exhaled volatile organic compounds. This review summarises and critically appraises the literature and offers suggestions for further research to apply exhaled biomarker analyses in clinical practice.

## 1. Background

Interstitial lung diseases (ILDs) are a heterogeneous group of disorders that are grouped together because they share clinical, radiographic, and pathologic features. ILDs are also referred to as diffuse parenchymal lung diseases (DPLDs) to clarify that the interstitium is not the only compartment of the lung affected.

The epidemiology of ILDs is characterised by significant heterogeneity, with wide variations in incidence and prevalence across different age groups, sexes, ethnicities, and geographical regions. For idiopathic pulmonary fibrosis (IPF), which predominantly affects older males (i.e., >60 years), a comprehensive analysis by Maher et al. [1] over 22 studies covering 12 countries revealed marked geographical variance. Adjusted incidence estimates (per 100,000 population) ranged from 0.9–4.9 in Europe and 3.5–13.0 in the Asia–Pacific region to a higher range of 7.5–9.3 in North America, whereas adjusted prevalence estimates (per 100,000) showed similar variability, ranging from 3.3–25.1 in Europe and 5.7–45.1 in Asia–Pacific countries to 24.0–29.8 in North America. In stark contrast to the male predominance in IPF, other common forms of ILD, such as idiopathic non-specific interstitial pneumonia (NSIP) and connective tissue disease-associated ILD (CTD-ILD), demonstrate a clear female predominance, with over half of cases occurring in middle-aged or older women [2,3]. While data on other individual ILDs are sparse, existing evidence suggests they are considerably less common than IPF [4]. Cumulatively, the overall prevalence for the entire ILDs spectrum is estimated to range from 6.3 to 76.0 cases per 100,000 people [4]. Finally, the interpretation of these epidemiological data is subject to significant caveats. These include inconsistencies in disease coding, which often fail to map precisely onto distinct clinical entities; methodological heterogeneity across studies; and disparities in healthcare access and the utilisation of electronic health records between different territories.

A significant number of ILDs are of unknown aetiology; consequently, they are classified under the umbrella term idiopathic interstitial pneumonia (IIPs) [5]. However, in many cases, specific aetiologic agents have been identified. Long-term exposure to occupational or environmental agents can have a toxic effect on the lungs [6]. Common agents are mineral dusts, organic dusts, and toxic gases. Many different types of mineral dust have correlations, but the ones frequently cited with the disease are silica, asbestos, coal mine dust, beryllium, and hard metal [7]. Organic dust includes mould spores and aerosolised bird droppings [8]. Inhaled toxic gases (methane, cyanide) affect the airways either by direct injury or through reactive oxygen molecules [9].

Exposure to tobacco smoke is another major risk factor for the development of ILDs [10,11]. Similarly, over 350 pharmacological agents have been implicated in the development of pulmonary complications, acting either through the formation of reactive metabolites or as part of a broader systemic inflammatory response [12,13]. Finally, systemic conditions such as connective tissue diseases and vasculitides can affect multiple compartments of the lung, including the bronchioles, parenchyma, and alveoli. For this reason, interstitial lung disease is a frequent manifestation of rheumatologic disorders [14].

Irrespective of the specific aetiology, these conditions converge upon a common pathogenetic pathway. The core pathobiological processes involve varying degrees of inflammation and fibrosis, which progressively disrupt the lung’s architecture [15]. This architectural distortion thickens the alveolar–capillary membrane, fundamentally impairing gas exchange and leading to the cardinal clinical manifestations of progressive exertional dyspnoea and a persistent, often dry, cough [16,17]. The consequent functional decline significantly diminishes quality of life and, in many cases, leads to death [18].

International guidelines, developed and periodically updated by leading respiratory societies such as the American Thoracic Society (ATS) and the European Respiratory Society (ERS), provide a structured approach to categorizing these disorders based on a combination of aetiological, clinical, radiological, and pathological features [5]. Broadly, ILDs are categorised into three major groups by their morphologic pattern: interstitial disorders, alveolar filling disorders, and combined/unclassifiable disorders. Each pattern could have a primary/idiopathic aetiology or a secondary aetiology when it is attributable to known causes or associations, including those linked to systemic autoimmune diseases (connective tissue diseases, or CTDs) [14], to occupational or environmental exposures to organic or inorganic dusts (e.g., hypersensitivity pneumonitis, asbestosis, silicosis) [6], or to certain medications [13] and radiation therapy [19]. There are also disorders predominantly affecting smokers [10], such as respiratory bronchiolitis-associated interstitial lung disease (RB-ILD) and alveolar macrophage pneumonia (AMP) [11]. Finally, a large and particularly challenging subset of ILDs are the idiopathic ones, for which no identifiable cause can be found. Among these, IPF is the most common and most lethal form [20].

## 2. Current Diagnostic Techniques and Their Limitations

The cornerstone of the modern diagnostic approach to ILD is the multidisciplinary team (MDT) discussion [21]. This collaborative process, which integrates the expertise of pulmonologists, radiologists, and pathologists, involves a comprehensive review of a patient’s clinical presentation, radiologic imaging, and, when available, histopathology from a lung biopsy. The MDT is considered the “gold standard” for diagnosis as it has been shown to improve diagnostic accuracy, enhance inter-observer agreement, and increase diagnostic confidence [22]. However, this gold standard is not without its flaws. The MDT is a resource-intensive process, often confined to specialised ILD centres, which creates significant disparities in access to high-quality care and can contribute to diagnostic delays [21,23]. Patients are frequently misdiagnosed with more common conditions such as asthma, chronic obstructive pulmonary disease (COPD), or congestive heart failure, leading to the empirical prescription of ineffective therapies and a cascade of unnecessary investigations.

High-resolution computed tomography (HRCT) is the essential imaging modality in the ILD workup [20]. In the appropriate clinical context, the identification of a “definite” Usual Interstitial Pneumonia (UIP) pattern on HRCT-characterised by subpleural, basal-predominant reticulation and honeycombing is highly specific for IPF and can obviate the need for a more invasive lung biopsy [24]. However, a large proportion of HRCT scans do not show this classic pattern. Instead, they reveal findings categorised as “probable UIP,” “indeterminate for UIP,” or suggestive of an alternative diagnosis, leaving significant diagnostic uncertainty.

When HRCT findings are not definitive, a surgical lung biopsy (SLB) is often considered the next step to secure a histopathological diagnosis [25]. While SLB can provide crucial diagnostic information, it is an invasive procedure burdened by considerable risks. Postoperative complications are common; 19.1% experienced one or more complications of surgery, including prolonged air leaks, pneumonia, and acute exacerbation of ILD [26]. The associated mortality risk is not trivial, estimated at approximately 4.4% [26]. Consequently, many patients, particularly those who are elderly or have severe physiological impairment (e.g., low diffusing capacity, pulmonary hypertension), are deemed unfit for SLB [27]. These patients are often left with an “unclassifiable” ILD, which complicates prognostic counselling and therapeutic decision-making.

Once an ILD diagnosis is established, the clinical focus shifts to monitoring for disease progression and assessing the response to therapy [20]. This process relies heavily on serial physiological measurements, which, while foundational to current practice, have significant limitations.

Pulmonary function tests (PFTs) represent a fundamental component of the diagnostic and management framework for ILDs, providing a non-invasive means to assess disease severity, establish prognosis, and monitor disease progression and response to therapy. Standard assessment comprises spirometry, the measurement of lung volumes, and diffusing capacity. Serial measurements of forced vital capacity (FVC) and diffusing capacity for carbon monoxide (DLCO) are used to track disease trajectory. A clinically meaningful decline in these parameters is a marker of disease progression and a predictor of increased mortality risk [28,29]. Despite their central role, PFTs are an imperfect tool. Their results are highly dependent on patient effort and technique, leading to considerable test-to-test variability that can obscure true changes in lung function [30]. More critically, PFTs lack sensitivity for detecting early or subtle disease. It is well established that patients can have substantial interstitial abnormalities on HRCT while maintaining PFT results within the normal range [28].

The six-minute walk test (6MWT) is a tool used to demonstrate oxygen requirements while also providing an assessment of a patient’s functional exercise capacity [20]. Desaturation is a strong predictor of mortality in ILD, particularly in IPF [31]. However, the 6MWT is inherently non-specific; performance can be significantly confounded by a host of non-pulmonary factors, including cardiovascular comorbidities, peripheral vascular disease, musculoskeletal pain (a particularly relevant issue in CTD-ILD), obesity, deconditioning, and patient motivation.

Significant limitations in the current diagnostic and monitoring pathway for ILDs create an urgent clinical imperative for the development of novel biomarkers. An ideal biomarker would be non-invasive, safe for serial assessment, cost-effective, and critically, highly sensitive to subtle changes in early disease activity. The clinical utility of such a tool would be profound: enabling earlier diagnosis, improving risk stratification for disease progression, providing objective endpoints for clinical trials, and guiding personalised therapeutic strategies in an era of expanding treatment options. In this context, the analysis of exhaled breath—a field known as “breathomics”—has emerged as a highly compelling and innovative approach to addressing this profound unmet need [32,33,34]. The contents of exhaled breath are diverse, ranging from small inorganic gas molecules (e.g., nitric oxide) to large, non-volatile particles like microbes and cytokines. Within this complex matrix, volatile organic compounds (VOCs) are of particular interest for monitoring health status. Their collective profile the “volatilome” provides a dynamic reflection of the body’s real-time metabolic state. This review will focus on the most common exhaled biomarker collection and analytical methods, such as the measurement of biomarkers in exhaled breath condensate (EBC) and the analysis of exhaled VOCs and exhaled nitric oxide (Table 1).

## 3. Technical Aspects of Exhaled Breath Analyses with a Focus on ILD

### 3.1. Exhaled Breath Condensate

The ERS technical standard document has summarised the most important technical aspects of EBC collection and analysis [32]. Collection material, temperature, and breathing pattern were identified as key factors influencing the composition of EBC in general; however, specific validation for each specific biomarker was suggested [32]. pH is the most commonly analysed biomarker in EBC. In patients with systemic sclerosis, low EBC pH at baseline was related to impairment in DLCO at 4 years follow up, indicating that it could be useful to predict disease progression [35]. However, the clinical value of EBC pH is limited by the fact that changes in environmental temperature and humidity [36,37], as well as simple physiological interventions, such as consuming food or beverages [38] or physical exercise [39,40] can significantly affect its value. Lower levels of EBC hydrogen peroxide ($H_{2}O_{2}$) were reported in ILD compared to obstructive lung diseases [41]. However, these differences could be due to variances in the breathing pattern rather than representing reduced levels. In addition, eating and environmental contamination [42], as well as diurnal variability [43], can affect EBC $H_{2}O_{2}$ levels, limiting its applicability to assess pulmonary oxidative stress in ILD. Concerning non-volatile molecules in EBC, very small or no differences were detected between patients with ILD and controls, as well as within ILD [44,45], with the exemption of the article by Rolla et al., reporting significantly higher EBC cytokine levels in patients with ILD compared to healthy controls [46]. For non-volatile compounds in EBC, the variable and unpredictable rate of dilution by the alveolar water loss is a major problem [47] that is very difficult to control [48]. This could explain why EBC results in ILD were not replicated by independent studies.

### 3.2. Exhaled Volatile Organic Compounds

When interpreting exhaled VOC results in ILD, it is important to notice if the analyses were focusing on individual specific compounds or their mixture using electronic noses [32]. Whilst the former has its merit in that the molecule, its biology, and its kinetics can be well characterised, single VOCs rarely represent complex pathologies. Analysing the whole mixture of VOCS (frequently termed as “breathprint”) could therefore be of better value in clinical practice than the analysis of a single molecule; this is true of the use of breathprints for both diagnostic and prognostic purposes [49]. Importantly, rigorous statistical scrutiny is warranted when comparing breathprints, as significantly different results are obtained depending on the statistical methods [50,51]. Standardisation of collection-related factors is emphasised by the ERS technical standard [32]. Expiratory flow rate and breath hold can significantly affect VOC concentration in exhaled breath samples when they are collected during a single exhalation [52]. Unfortunately, this has not always been controlled in ILD studies [53,54,55,56]. Reassuringly, recent large-scale studies standardised these parameters in patients with various forms of ILD [57,58,59]. Breathing pattern can affect exhaled VOC levels when the samples are obtained by multiple breaths [60]. Breathing pattern has been controlled in the most recent studies [61,62], but not in the earlier works [63,64,65]. Some exhaled VOCs are affected by age, sex [66,67], smoking, and environmental pollution [68]; therefore, it is important to carefully select controls for patients with ILD, as well as sites for multi-centre studies. Furthermore, diet [52,69,70] and physical exercise [39,40] may change the concentration of exhaled VOCs; therefore, it is a best practice to avoid these before sampling and to record them in the clinical research forms. Finally, exhaled VOCs shows significant diurnal variations [71], contributing to a further layer of complexity when interpreting data in ILD. Finally, volatile metabolites arising from medications can significantly alter the exhaled VOC pattern [72,73]. Different exhaled VOC profiles were also associated with hypoxia as well as hyperoxia [74]. Further studies are warranted to investigate the effect of medications and oxygen therapy in ILD [57,58,59]. Patients with ILD often suffer from chronic comorbidities. Exhaled VOC profiles were reported to be different in diseases of the heart [75], disease of the kidney [76], diabetes [77], or reflux disease [78]. Controlling for comorbidities in clinical studies is therefore warranted.

**Table 1 diagnostics-15-02884-t001:** Comparison of three major methods of exhaled breath analysis in ILDs.

	Principles	Strengths	Limitations
Exhaled Breath Condensate (EBC)	Involves the collection and analysis of biomarkers within the condensed liquid from exhaled breath. Common biomarkers analysed include pH, hydrogen peroxide ($H_{2}O_{2}$), and non-volatile molecules like cytokines.	Some studies suggest potential clinical value for specific biomarkers (pH, cytokines).	The clinical value is considered limited. Measurements are highly susceptible to confounding factors, including environmental temperature/humidity, food and drink consumption, physical exercise, breathing patterns, and diurnal variability.
Exhaled Volatile Organic Compounds (eVOCs)	Involves analysing volatile organic compounds (VOCs), which provide a dynamic reflection of the body’s real-time metabolic state. Analysis can target individual specific compounds or, more commonly, the entire mixture of VOCs (the “breathprint”) using technologies like electronic noses.	Analysing the “breathprint” is considered to have better clinical value than analysing single molecules. Electronic nose technology has shown superior discriminatory ability in distinguishing ILD patients from healthy controls and from other diseases like COPD, asthma, and lung cancer. It has also shown capacity to discriminate between ILD subtypes (e.g., IPF vs. non-IPF) and may have prognostic value, with VOCs correlating to changes in FVC, DLCO, and survival.	Single VOCs rarely represent complex pathologies. The method is highly sensitive to a vast number of confounders, including collection methods (expiratory flow rate, breath hold), breathing patterns, patient demographics (age, sex), lifestyle factors (smoking, diet, exercise), environmental pollution, diurnal variations, medications, oxygen therapy, and chronic comorbidities (e.g., heart disease, diabetes, reflux). Results are also highly dependent on the statistical methods used.
Exhaled Nitric Oxide	Measures the concentration of nitric oxide (NO) in exhaled breath. Extended NO analyses (using multiple flow rates) can partition NO production to the central airways (measured as $Jaw_{NO}$) or the peripheral/alveolar region (measured as $CA_{NO}$).	$CA_{NO}$ has been found to be increased in some ILD patients (especially CTD-ILD) and showed an inverse relationship with DLCO. It has also shown potential in predicting treatment response.	$FE_{NO}$ is affected by age, sex, height, weight, smoking, medications, and diet.

### 3.3. Exhaled Nitric Oxide

The ATS, together with the ERS, issued their recommendations for exhaled nitric oxide measurements in 2005 [79]. The ATS has later published guidelines to help interpreting fractional exhaled nitric oxide (FE_NO_) results [80]. These documents, together with the ERS technical standard, have acknowledged that FE_NO_ is affected by age, sex, height and weight, smoking, medications, and diet [32,79,80]. Therefore, similarly to exhaled VOCs, in studies on exhaled nitric oxide, including FE_NO_, CA_NO_ (concentration of nitric oxide in the gas phase of the alveolar or acinar region) and Jaw_NO_ (total flux of NO in the conducting airway compartment), ILD groups should have had appropriate controls. Whilst FE_NO_ is commonly used in clinical practice, the measurement of exhaled nitric oxide also allows for the partitioning of nitric oxide production at the central and peripheral/alveolar parts of the lung [32]. This is particularly relevant in ILD, where the inflammation occurs in the peripheral/acinar airways and alveoli, which are parts of the lung that are difficult to investigate. Tiev et al. and Girgis et al. reported an inverse relationship between CA_NO_ and DLCO in patients with scleroderma [81,82,83]. Similar results were later reported for CTD-ILD, IPF, NSIP [84], and sarcoidosis [85], suggesting that extended NO analyses may have value in understanding the site of inflammation in ILD. Whilst the technique is promising for clinical practice, patients with severe lung disease may struggle to produce acceptable readings at both very low and very high flow rates [86].

## 4. Summary of Studies in ILD

The presence of elevated levels of VOCs resulting from oxidative stress, such as benzaldehyde, dimethylsulfide, dimethylsulfone, and 2,5-dimethylfuran [53], as well as collagen-derived components, including alanine, proline, valine, leucine/isoleucine, and 4-hydroxyproline [87], supports the pathophysiological hypothesis of inflammation and increased collagen turnover in patients with ILDs. However, to date, no single VOC has proven to be a reliable diagnostic biomarker for ILDs. This also applies to nitric oxide in exhaled breath, which, after its discovery in the late 1990s, attracted significant attention because its production in the respiratory tract can increase due to an inducible NO synthase (iNOS) during inflammation and cellular repair and healing processes [88]. The hypothesis that an increase in FE_NO_, which can be easily detectable, could be used to distinguish patients with ILDs from healthy controls and other respiratory diseases led to numerous publications in the following years, with conflicting results. The studies by Zheng et al. [89] and Krauss et al. [45] showed no discriminatory power of FE_NO_ measured at a flow rate of 50 mL · $s^{-1}$ between ILD, healthy controls, and other chronic respiratory diseases such as COPD and lung cancer, nor between ILD subtypes, while others showed results only in certain specific cases. Guilleminault et al. reported a higher median FE_NO_ in 13 patients with chronic hypersensitivity pneumonitis (HP) compared to IPF, CTD-ILD, and drug-induced ILD (DIILD), with good discriminatory capacity (i.e., ROC-AUC 0.85), while Ferrer-Pargada found a higher median FE_NO_ in patients with COVID-19 pneumonia who later developed sequelae [90]. In general, however, FE_NO_ values measured at 50 mL · $s^{-1}$ in patients with ILDs are around 25 ppb, with differences that—although statistically significant in some studies—are not clinically relevant. One explanation for this apparent discrepancy is that FE_NO_ at 50 mL · $s^{-1}$ reflects NO production in the airways rather than in the alveoli, which are the main site of inflammation and fibrogenesis in ILDs [88]. For this reason, some studies have analysed NO production at different flow rates, allowing for the measurement of Jaw_NO_ and CA_NO_. CA_NO_ was found to be increased in patients with ILD, particularly in CTD-ILD and in forms secondary to systemic sclerosis, with a negative correlation with total lung capacity and DLCO values [81,82,84]. However, this finding has not been consistently confirmed across all studies [91,92], and its discriminatory capacity was acceptable only for CTD-ILD forms [84,93], in which breath analysis also showed potential in predicting treatment response [94]. Few studies have analysed compounds in EBC, such as prostaglandins, eicosanoids, nitrites, and nitrates [35,45], but without significant results, except for an association between reduced pH and functional decline and death [35]. A detailed analysis of the studies on exhaled nitric oxide and exhaled breath condensate is reported in Table 2.

Given the limited results obtained from the analysis of single volatile compounds, diagnostic approaches have therefore relied on either groups of VOCs or on analysis of the entire spectrum of compounds (“breathprint”) using electronic nose technology (Figure 1). The first approach employed mass spectrometry for VOC identification and achieved a receiver operating characteristic area under the curve (ROC-AUC) for distinguishing ILD patients from healthy controls ranging from 80% to 90% [53,87]. The latter demonstrated superior discriminatory ability, with ROC-AUC values between 90% and 100%, independent of the type of electronic nose used [57,58,59,64]. Importantly, these studies also investigated the capacity for differential diagnosis against other diseases such as COPD, asthma, and lung cancer—an aspect not assessed in studies using groups of VOCs. The results were positive, with ROC-AUC values in testing cohorts of 0.98 (95% CI 0.94–1.00) for lung cancer, 1.00 (95% CI 1.00–1.00) for asthma, and 0.96 (95% CI 0.90–1.00) for COPD [59]. Given the shared smoking exposure and overlapping clinical features between some ILDs and COPD, the latter has been the most extensively studied differential diagnosis. Positive results have been confirmed by Dragonieri et al. [56], who reported a 96.7% discriminatory accuracy between IPF, COPD, and healthy controls in an external validation cohort, and by Krauss et al. [64], who found ROC-AUC values of 0.91 versus HP, 0.77 versus cryptogenic organising pneumonia, and 0.85 versus CTD-ILD. None of these studies, however, performed a sensitivity analysis for smoking exposure, which differed between groups and may represent a confounding factor.

ILDs are a heterogeneous group of disorders, and classification according to current guidelines is essential for appropriate management and treatment. A key distinction is between IPF and non-IPF forms, particularly CTD-ILD, as these are more frequently associated with progressive fibrosing phenotypes [97]. Gas chromatography studies have shown that VOC group analysis can discriminate between these two conditions, with an ROC-AUC of approximately 85% in testing cohorts [53,54]. Breathprint analysis results are consistent with these findings, demonstrating a discriminatory capacity between IPF and non-IPF ILDs of 0.87 (95% CI 0.77–0.96) in the testing cohort, with non-cross-validated diagnostic accuracy ranging from 77% for IPF versus HP to 94% for IPF versus CTD-ILD [57]. These ROC-AUC values are slightly higher than those reported by Krauss et al. [64]. Of particular interest is the ability to distinguish sarcoidosis from healthy subjects (ROC-AUC 1.00), regardless of pulmonary or multiorgan involvement or ongoing immunosuppressive therapy, as well as from other ILDs, with a testing ROC-AUC of 0.87 (95% CI 0.82–0.93) [58]. Similarly, pneumoconiosis has been discriminated with an ROC-AUC of 0.86, independent of sex and smoking status [98]. More recently, van der Sar et al. addressed the challenge of DIILDs, particularly those related to anticancer therapies, which represent an emerging diagnostic challenge. In their pilot study of 40 subjects (20 with multidisciplinary team-diagnosed DIILD), exhaled breath analysis yielded a testing ROC-AUC of 0.81 (95% CI 0.67–0.95) [99]. Another diagnostic challenge is identifying, among systemic sclerosis patients, those who will develop interstitial lung involvement. Massenet et al. [65] demonstrated that using nine VOCs, it is possible to discriminate systemic sclerosis patients with ILD from those without, with a non-cross-validated ROC-AUC of 0.82-comparable to that obtained with DLCO analysis [65]. This finding is consistent with Marges et al., who, in a larger cohort of 110 SSC-ILD and 113 SSc-non-ILD patients, reported a testing ROC-AUC of 0.84 (95% CI 0.75–0.94), with discriminatory capacity independent of disease duration, severity, and immunosuppressive treatment [100]. However, it should be noted that the study was not powered to detect these latter effects.

Given the potential for sudden and unpredictable disease progression in ILDs, with significant prognostic implications, studies have begun to investigate associations between VOCs and functional parameters of pulmonary deterioration. In a cohort of 57 IPF patients, VOC group analysis was associated with longitudinal changes in FVC and DLCO, as well as with survival [62]. This is consistent with the notion that exhaled breath analysis reflects underlying pulmonary pathophysiology, as evidenced by the correlation between total bronchoalveolar lavage cell counts and breath analysis results [56]. A detailed analysis of studies on exhaled volatile organic compounds is reported in Table 3.

## 5. Summary of Evidence, Clinical Implications and Agenda for Future Research

The exploration of exhaled biomarkers in ILD represents a rapidly evolving frontier in respiratory medicine. Traditional diagnostic and monitoring tools, including HRCT, PFTs, and MDT assessment, remain indispensable, but their limitations are well recognised. HRCT is often inconclusive, surgical lung biopsy carries substantial risk, and physiological parameters such as FVC lack the sensitivity to detect early disease progression. Against this backdrop, exhaled breath analysis, encompassing VOCs, EBC, and nitric oxide fractions, emerges as a promising, non-invasive modality that may complement existing strategies.

### 5.1. Summary of Evidence

The cumulative body of work demonstrates that exhaled VOCs reflect core pathophysiological processes of ILDs, including oxidative stress and collagen turnover. Elevated levels of aldehydes, sulfides, and collagen-derived amino acids in patients with IPF and other ILDs support this mechanistic link. However, single compounds have not proven reliable as standalone biomarkers. Instead, discriminatory capacity has been achieved through composite “breathprints” or panels of VOCs. Electronic nose technologies, when applied in large multicentre studies, have distinguished ILDs from healthy controls and from other respiratory diseases with remarkable accuracy, often yielding ROC-AUC values approaching or exceeding 0.95. Beyond diagnosis, breathomics has demonstrated capacity for disease stratification. Distinction between IPF and non-IPF ILDs, including CTD-ILD, HP, and sarcoidosis, has been shown feasible, with diagnostic accuracies in the range of 77–94%. Importantly, early data suggest prognostic associations, as specific VOC signatures correlate with longitudinal declines in FVC, diffusing capacity, and survival. Comparable findings have been reported for systemic sclerosis, in which breath analysis could discriminate patients with subclinical interstitial involvement. Parallel investigations into EBC and FE_NO_ provide further—though less consistent—evidence: alveolar nitric oxide levels inversely correlate with DLCO in various ILDs. Despite these advances, substantial methodological heterogeneity persists. Studies vary widely in sampling techniques, analytic platforms, and statistical approaches, while patient cohorts often differ in age, smoking history, environmental exposures, and comorbidities—factors known to influence breath composition. These sources of variability explain, in part, why promising findings have not consistently translated across studies or reached clinical adoption.

### 5.2. Clinical Implications

The potential clinical applications of exhaled breath analysis in ILD are manifold. First, as a diagnostic adjunct, breathomics may reduce reliance on invasive biopsy in indeterminate cases, particularly when HRCT findings are inconclusive. Second, for prognostication, breath-derived biomarkers may enable earlier identification of patients at risk of rapid disease progression, thereby guiding timely therapeutic intervention. Third, exhaled biomarkers may serve as non-invasive endpoints in clinical trials, offering sensitive measures of treatment response that transcend the limitations of spirometry. Finally, the feasibility of repeated, safe, and inexpensive sampling positions exhaled breath analysis as an attractive tool for longitudinal monitoring in routine practice. Nevertheless, enthusiasm must be tempered by recognition of current limitations. Breath analysis is not yet standardised; variability in collection devices, expiratory flow patterns, environmental contamination, and statistical modelling all influence results. Furthermore, comorbid conditions common in ILD populations, such as cardiovascular disease, gastro-oesophageal reflux, and diabetes, have been shown to alter exhaled VOC profiles. Without careful control, these factors may confound interpretation and reduce specificity.

### 5.3. Agenda for Future Research

Future investigations must prioritise methodological rigor and standardisation. The adoption of technical standards, such as those proposed by the ERS for EBC and VOC analysis, is imperative. Harmonisation across centres will facilitate reproducibility, meta-analysis, and eventual regulatory acceptance. Large, prospective, multicentre cohorts with standardised protocols are required to validate discriminatory and prognostic breathprints across diverse ILD phenotypes. In parallel, integration of breathomics with genomic, proteomic, and imaging data offers an avenue toward multi-modal biomarker signatures that more comprehensively capture disease biology. Mechanistic research is equally crucial. Establishing causal links between identified exhaled compounds and underlying fibrotic or inflammatory pathways will bolster biological plausibility and aid interpretation. Studies should also evaluate the impact of common confounders, including diet, circadian variation, oxygen supplementation, and pharmacotherapy, on exhaled biomarker profiles. Finally, the incorporation of artificial intelligence and advanced machine learning methods may enhance the robustness of pattern recognition and facilitate translation into point-of-care diagnostics.

## 6. Conclusions

Exhaled breath analysis in ILD has matured from exploratory pilot studies to large, rigorously conducted trials that demonstrate strong diagnostic and prognostic potential. This approach offers an unprecedented opportunity to overcome the limitations of current diagnostic pathways by providing non-invasive, repeatable, and physiologically relevant biomarkers. However, before exhaled biomarkers can be integrated into clinical practice, further validation, standardisation, and mechanistic understanding are essential. With these refinements, breathomics may evolve into a transformative tool for precision medicine in ILD, reshaping both research frameworks and clinical care.

## Figures and Tables

**Figure 1 diagnostics-15-02884-f001:**
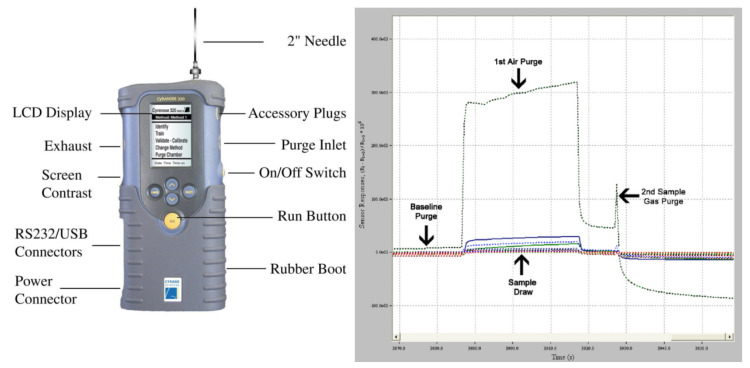
Cyranose 320 device (**left**) and example output (**right**).

**Table 2 diagnostics-15-02884-t002:** Studies on exhaled nitric oxide and exhaled breath condensate.

Study	Exhaled Biomarker	Study Population	Outcome	Ref.
Cameli et al., 2019	FE_NO_ at multiple flow-rates (50–100–150 and 350 mL · $s^{-1}$), CA_NO_ and Jaw_NO_.	ILD: 134 (IPF: 50, NSIP: 19, HP: 19, CTD-ILD: 46), HC: 60	CA_NO_ and FE_NO_ 150 and 350 mL · $s^{-1}$ were significantly higher in ILD patients (median ≥ 10 ppb) than HC (5 ppb), with the exception of FE_NO_ 350 mL · $s^{-1}$ in HP (−8 ppb) vs. HC. CA_NO_ was inversely correlated with FVC and DLCO. ROC-AUC for CA_NO_ in discriminating CTD-ILD from all others was 0.795, with a optimal cut-point of 13 ppb.	[84]
Guilleminault et al., 2013	FE_NO_ at 50 mL · $s^{-1}$	ILD: 61 (IPF: 18, CTD-ILD: 22, HP: 13, DIILD: 8)	Median FE_NO_ was 51 ppb (IQR 36-74) in HP, significantly higher than other ILDS. ROC-AUC in discriminating HP from other ILDs was 0.85.	[90]
Ferrer-Pargada et al., 2025	FE_NO_ at 50 mL · $s^{-1}$	COVID-19 pneumonia with oxygen requirements and need of follow-up for diffuse interstitial lung disease: 335	FE_NO_ was higher in patients with interstitial lung sequelae (median 24 ppb) than in patients without (median 20 ppb). ROC-AUC in discriminating patients with lung sequelae was 0.63 (95%CI 0.57–0.69).	[95]
Girgis et al., 2002	FE_NO_ at multiple flow-rates (50–100–150 and 200 mL · $s^{-1}$), CA_NO_ and Jaw_NO_.	SSc: 20 (with ILD: 15, with PH: 5), HC: 20	CA_NO_ was increased and Jaw_NO_ was reduced in SSc versus HC. There was a negative correlation between CA_NO_ and DLCO among the patients with SSc (r = −0.66).	[81]
Kozij et al., 2017	FE_NO_ at multiple flow-rates (50–100–150–200 and 250 mL · $s^{-1}$), CA_NO_ and Jaw_NO_.	SSc: 35 (Without ILD and PH: 16, With ILD and without PH: 12, With ILD and PH: 7), HC: 25, SLE-PH: 6, Idiopathic PH: 9	Jaw_NO_ was reduced in ILD patients (median 1009 ppb) than in HC (median 1342 ppb), without significant differences in CA_NO_.	[91]
Oishi et al., 2017	FE_NO_ at 50 mL · $s^{-1}$ and CA_NO_.	Acute onset ILD: Eos. pneumonia: 18, COP: 16, Sarcoidosis: 5, HP: 3	FE_NO_ value of patients with EP (48.1 ppb) was significantly higher than that of the other groups. ROC-AUC in discriminating Eos.pneumonia from other ILDs was 0.90 for FE_NO_ and 0.85 for CA_NO_.	[96]
Zheng et al., 2021	FE_NO_ at 50 mL · $s^{-1}$	ILD: 95 (Dermatomyositis-assoc. CTD-ILD: 69, Sjögren’s-assoc. CTD-ILD: 7, Mixed CTD-ILD: 9, IPF: 5, HP: 5), CTD without ILD: 82, HC: 24	FE_NO_ did not significantly differ between ILD and HC and among ILD subgroups.	[89]
Wuttge et al., 2010	CA_NO_	Early onset SSc: 34, HC: 26	CA_NO_ was higher in patients with SSc (median 3–3.5 ppb) versus HC (median 2 ppb), but did not differ between SSc with and without ILD. CA_NO_ did not correlate with pulmonary function tests.	[92]
Tiev et al., 2009	FE_NO_ at multiple flow-rates (50–100–150 and 200 mL · $s^{-1}$) and CA_NO_.	SSc: 65 (with ILD: 38, without ILD: 27)	CA_NO_ is higher in SSc with ILD. A cut-off level of 4.3 ppb has a sensitivity of 87% and a specificity of 59% in discriminating SSc with from without ILD.	[93]
Tiev et al., 2007	FE_NO_ at multiple flow-rates (50–100–150 and 200 mL · $s^{-1}$) and CA_NO_.	SSc: 58 (with ILD: 33, without ILD: 25), HC: 19	CA_NO_ was higher in SSc versus HC, and higher in SSc with ILD (median 7.5 ppb) versus without ILD (median 4.9 ppb). CA_NO_ was inversely correlated with TLC (r = −0.34) and DLCO (r = −0.37).	[82]
Tiev et al., 2014	FE_NO_ at multiple flow-rates (50–100–150 and 200 mL · $s^{-1}$) and CA_NO_.	SSc treated with 6 courses of cyclophosphamide: 19	6 out of 7 patients with baseline CA_NO_ > 8.5 ppb had an improvement in FVC or TLC > 10% from baseline versus 3 out of 12.	[94]
Krauss et al., 2019	FE_NO_ at 50 mL · $s^{-1}$, PGE2 and 8-Isoprostan in EBC and BALF.	ILD: 34 (IPF: 11, RB-ILD: 2, COP: 8, HP: 5, Sarcoidosis: 3, CTD-ILD: 3, Indeterminate ILD: 2), COPD: 24, Lung cancer: 16, HC: 20	No meaningful differences between FE_NO_ or eicosanoid values in EBC and BALF of the different cohorts as well as HC.	[64]
Guillen-Del Castillo et al., 2017	FE_NO_ at 50 mL · $s^{-1}$, exhaled CO, pH, nitrite, nitrate and IL-6 in EBC. PFTs performed annually for 4 years.	SSc: 35 (with ILD: 12, without ILD: 23)	The pH and FE_NO_ were lower in patients showing a functional decline or death. The ROC-AUC in predicting functional decline or death was 0.65 (95%CI 0.41–0.89) for pH, and 0.81 (95%CI 0.65–0.96) for FE_NO_. No difference for other substances.	[35]

Legend: AUC: area under the receiver operating curve; BALF: bronchoalveolar lavage fluid; CA_NO_: alveolar concentration of NO; COP: cryptogenic organising pneumonia; COPD: chronic obstructive lung disease; CTD-ILD: connective tissue disease-associated ILD; DIILD: drug-induced interstitial lung disease; DLCO: diffusing capacity of the lungs for carbon monoxide; EBC: exhaled breath condensate; FVC: forced vital capacity; HC: healthy controls; HP: hypersensitivity pneumonitis; ILD: interstitial lung disease; IPF: idiopathic pulmonary fibrosis; Jaw_NO_: maximum airway flux of NO; PH: pulmonary hypertension; RB-ILD: respiratory bronchiolitis ILD; SLE: systemic lupus erythematosus; SSc: systemic sclerosis; TLC: total lung capacity.

**Table 3 diagnostics-15-02884-t003:** Studies on exhaled volatile organic compounds.

Study	Study Population	E-Nose/Type of Discriminant Analysis	Outcome	Ref.
Van der Sar et al., 2023	ILD: 161, Asthma: 65, COPD: 50, Lung cancer: 46	SpiroNose. Training and testing (PLS-DA) set. Training and testing groups were obtained using function “sample” in R.	ROC-AUC in discriminating ILD from all other diseases of 0.99 (95% CI 0.97–1.00) in the test set. AUC of 1.00 (95% CI 1.00–1.00) for asthma, AUC of 0.96 (95% CI 0.90–1.00) for COPD, and AUC of 0.98 (95% CI 0.94–1.00) for lung cancer in test sets.	[59]
Dragonieri et al., 2020	Training: IPF: 32, COPD: 33, HC: 36. Testing: IPF: 10, COPD: 10, HC: 10.	Cyranose320. Training (PCA+LDA) and testing on an independent validation cohort.	IPF vs. COPD vs. healthy controls: CVA 96.7% in external validation. There is a correlation between BALF total cell count and both Principal Components 1 and 2.	[56]
Krauss et al., 2019	ILD: 174 (COP: 28, IPF: 51, HP: 20, CTD-ILD: 25, Sarcoidosis: 19), COPD: 23, HC: 33	Aeonose. A software program called Aethena was used for pre-processing, data compression, and neural networking.	IPF vs. HC: AUC 0.95, MCC 0.73; COP vs. HC: AUC 0.89, MCC 0.67; CTD-ILD vs. HC: AUC 0.90, MCC 0.69. Other ILD were not discriminated from HC. IPF vs. COP: AUC 0.82, MCC 0.49; IPF vs. CTD-ILD: AUC 0.84, MCC 0.55; CTD-ILD vs. COP: AUC 0.75, MCC 0.40.	[64]
Van der Sar et al., 2022	Sarcoidosis: 252, ILD: 317 (IPF: 124, CTD-ILD: 64, HP: 50), HC: 48	SpiroNose. Random assignment (2:1) in training and testing (PLS-DA) set.	Sarcoidosis VS HC testing set: 1.00 (independent from sarcoidosis pulmonary involvement, multiple organ involvement, and immunosuppressive treatment). Sarcoidosis VS ILD testing set: AUC of 0.87 (95%CI, 0.82–0.93). Sarcoidosis VS HP testing set: AUC of 0.88 (95%CI, 0.75–1.00).	[58]
Van der Sar et al., 2023	Sarcoidosis: 252, ILD: 317	SpiroNose. Comparison of various statistical methods.	A classification model with feature selection and random forest classifier showed the highest accuracy (87.1%).	[51]
Van der Sar et al., 2024	DIILD: 20, HC: 20	SpiroNose. Random assignment (2:1) in training and testing (PLS-DA) set.	ROC-AUC DIILD vs. HC: 0.81 (95% CI 0.67–0.95). The ROC-AUC was higher in DIILD without corticosteroids (0.87) than in DIILD with corticosteroids (0.80).	[99]
Yang et al., 2017	Pneumoconiosis: 34, HC: 64	Cyranose 320. Random assignment (80%/20%) in training (LDA) and testing set.	Pneumoconiosis vs. HC testing set: AUC 0.86. Sensitivity and specificity were 66.7% and 71.4%, respectively. Results are not influenced by smoking and gender.	[98]
Moor et al., 2021	ILD: 322 (Sarcoidosis: 141, IPF: 85, CTD-ILD: 33, HP: 25, Idiopathic NSIP: 10, IPAF: 11, Other: 17), HC: 48	SpiroNose. PLS-DA, with a training and testing set obtained using function ’sample’ in R.	ILD vs. HC training and testing set: AUC 1.00. IPF versus non-IPF ILDs: training AUC 0.91 (0.85–0.96), testing AUC 0.87 (0.77–0.96).	[57]
Marges et al., 2024	SSc: with ILD: 110, Without ILD: 113	SpiroNose. PLS-DA, with a training and testing set (ratio 2:1).	SSc-ILD vs. SSc no ILD: training AUC 0.79 (0.72–0.87), testing AUC 0.84 (0.75–0.94). No impact of immunosuppressant use, disease duration and severity was found.	[100]
Yamada et al., 2017	IPF: 40, HC: 55	Multi-capillary column and ion mobility spectrometer.	Acetoin, p-cymene, isoprene, ethylbenzene and an unknown compound were significantly different in IPF than in HC, with the first being correlated with pulmonary function tests.	[101]
Gaugg et al., 2019	IPF: 21, HC: 21	Secondary electrospray ionisation–mass spectrometry (SESI-MS). 1 million 10-fold cross-validations.	Significantly elevated levels of alanine, proline, valine, leucine/isoleucine and 4-hydroxyproline in the EBC of IPF patients. ROC-AUC in discriminating IPF from HC: 0.84 (95%CI 0.78–0.88).	[87]
Plantier et al., 2022	ILD: 104 (IPF: 53, CTD-ILD: 51), HC: 51	Gas chromatograph time-of-flight mass spectrometry. Random Forest with training/testing set (80%/20%).	The AUC in discriminating IPF from HC, CTD-ILD from HC and IPF from CTD-ILD was 91.2%, 83.9%, and 83.8%, respectively. Positive correlation between VOCs and TLC and 6MWD.	[53]
Hayton et al., 2025	IPF: 57	Gas chromatography–mass spectrometry. LASSO regression model.	63 VOCs associated with a change in FVC and 28 with DLCO. VOCs associated with survival.	[62]
Massenet et al., 2024	SSc: with ILD: 21, Without ILD: 21	Gas chromatography high-resolution time-of-flight mass spectrometry. PLS-DA on 9 VOCs.	ROC-AUC in discriminating SSc-ILD vs. SSc no ILD: 0.82.	[65]
Taylor et al., 2024	IPF: 12, CTD-ILD: 13	Liquid chromatography–mass spectrometer. PLS-DA, with a training and testing set.	Testing ROC-AUC in discriminating IPF from CTD-ILD of 0.88. ROC-AUC in disease severity: 0.82 (DLCO), 0.90 (FEV1) and 0.87 (FVC), respectively.	[54]

Legend: AUC: area under the receiver operating curve; BALF: bronchoalveolar lavage fluid; COPD: chronic obstructive lung disease; CTD-ILD: connective tissue disease-associated ILD; DIILD: drug-induced interstitial lung disease; HC: healthy controls; HP: hypersensitivity pneumonitis; ILD: interstitial lung disease; IPAF: interstitial pneumonia with autoimmune features; IPF: idiopathic pulmonary fibrosis; MCC: Matthews’s correlation coefficient; LDA: linear discriminant analysis; PCA: principal component analysis; PLS-DA: partial least square discriminant analysis; SSc: systemic sclerosis.

## Data Availability

Due to privacy and ethical considerations, corresponding author can provide the dataset supporting the reported results upon request. Interested parties can contact corresponding author directly via email to request access.

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
