# Peer review of "The Role of Exhaled Breath Analyses in Interstitial Lung Disease"

_diagnostics, 2025, doi:10.3390/diagnostics15222884_

Round 1
Reviewer 1 Report
Comments and Suggestions for Authors
This is an interesting narrative review article discussing the clinical implications of exhaled breath analysis in interstitial lung diseases (ILDs). The authors explain in detail the three major methods, i.e., exhaled nitric oxide, exhaled breath condensate, and exhaled volatile organic compounds, and report relevant findings from previous studies in ILDs. The review comprehensively summarises the existing knowledge and can stand as a useful guide for the translational and clinical researcher in the field. While the article is of adequate quality, the authors are advised to consider the following minor additions or modifications:
1. The length of the Background section is a bit larger than expected as an introduction to the main part of the article. The current diagnostic techniques in ILDs and their limitations could be included in a brief separate section, just after the Background section, to emphasise further where we stand now and why we need additional tools to investigate ILDs.
2. The concept of multimodal biomarkers for ILDs in clinical practice is the key to address the importance of the article. The unmet need for novel biomarkers could be further underscored in the Background section with the inclusion of relevant references, e.g., Stainer et al. 2021 (DOI: 10.3390/ijms22126255) and Tomos et al. 2023 (DOI: 10.3390/biomedicines11102796).
3. The word “Conclusions” could be replaced by the phrase “Summary of evidence” in the title of the section “Conclusions, clinical implications and agenda for future research”, as there is already a Conclusion section at the end of the article.
4. A table or a figure synopsising the three major methods of exhaled breath analysis in ILDs and comparing their principles, strengths, and limitations would be useful to be included.
Reviewer 2 Report
Comments and Suggestions for Authors
This review discusses the utility of exhaled NO and VOC measurements in interstitial lung disease (ILD). The authors cite numerous studies related to VOCs in ILD and emphasize their potential usefulness.
However, the section on ILD feels somewhat lengthy and could be more concise. In addition, a new guideline has recently been published (Ryerson, C. J., et al., 2025. Update of the International Multidisciplinary Classification of the Interstitial Pneumonias: An ERS/ATS Statement. Eur Respir J.), and given the timing of publication, it would be appropriate to reference and incorporate it.
While VOC analysis may be relatively familiar to pulmonologists, surgeons, pathologists, and radiologists may be less acquainted with the technique. Including examples of the equipment used for VOC measurement and representative output data or spectra would help readers better visualize the method and its clinical relevance.
